# Rethinking Mental Automatism: De Clérambault’s Theory in the Age of Novel Psychoactive Drugs: Psychotropic Effects and Synthetic Psychosis

**DOI:** 10.3390/healthcare12121172

**Published:** 2024-06-10

**Authors:** Valerio Ricci, Giuseppe Maina, Giovanni Martinotti

**Affiliations:** 1Psychiatry Department, San Luigi Gonzaga Hospital, University of Turin, Regione Gonzole 10, 10043 Orbassano, Italy; giuseppe.maina@unito.it; 2Department of Neurosciences “Rita Levi Montalcini”, University of Turin, Regione Gonzole 10, 10043 Orbassano, Italy; 3Department of Neurosciences, Imaging and Clinical Sciences, Università degli Studi G. D’Annunzio Chieti-Pescara, 66100 Chieti, Italy; giovanni.martinotti@gmail.com

**Keywords:** novel psychoactive drugs, substance misuse, psychosis, mental automatism, schizophrenia, cannabis, De Clerambault

## Abstract

The widespread use of novel psychoactive substances (NPSs)—defined as new narcotic or psychotropic agents not classified under the Single Convention on Narcotic Drugs of 1961 or the Convention on Psychotropic Substances of 1971—poses a significant challenge to contemporary mental health paradigms due to their impact on psychiatric disorders. This study revisits and expands upon the theory of mental automatism as proposed by Gaëtan Gatian de Clérambault, aiming to elucidate the psychopathological mechanisms underlying substance-induced psychoses (SIP) and their distinction from non-induced psychoses (schizophrenia and related disorders). Through a phenomenological and clinical investigation, we explore the relevance of mental automatism in the development of toxic psychoses, drawing upon the historical and contemporary literature. This research highlights the psychopathological distinctions between induced and non-induced psychoses and the transition mechanisms from acute to chronic psychosis states. De Clérambault’s theory, supplemented by Janet, Jackson, and Bonhoeffer’s contributions, provides a foundational framework for understanding the genesis of SIP. Our findings suggest that NPS consumption, particularly among adolescents and psychiatric patients, significantly correlates with increased risks of SIP, marked by a transition to chronicity influenced by biological lesions triggered by substance use. Furthermore, we propose a comprehensive framework for SIP, integrating mental automatism, psychopathological distinctions, and transition mechanisms. This framework aims to refine diagnostic criteria and therapeutic approaches, addressing gaps in clinical practice and research. The study underscores the need for a nuanced understanding of SIP, advocating for a paradigm shift in psychiatric assessment and treatment approaches to better address the complexities of substance-induced mental health disorders.

## 1. Introduction

### 1.1. Background on Substance-Induced Psychosis

Legal and illegal psychoactive substances are readily accessible and affordable for contemporary adolescents. They offer quick and effortless alterations of consciousness and emotional states in response to the constantly changing environment. Substance use disorders (SUDs) represent a significant public health issue, as evidenced by their ranking among the primary contributors to years lived with disability globally in 2010 [1]. Presently, substance use has become pervasive to the extent that it can scarcely be categorized as deviant behavior, despite its severe implications for mortality rates and psychiatric burdens. Devereux [2] posited that each society establishes its own standards and regulations, which shape individuals’ expressions of both illness/distress and success/fulfillment. Within the framework of contemporary society, substance use intertwines with both of these aspects. Stimulants are often perceived as facilitating successful lifestyles, bolstering a sense of “empowerment” and augmenting the narcissistic invulnerability present in certain personality types. Conversely, substance use tends to engender a sense of completeness, which disregards intricacies and diversity in favor of a singular experience, thereby alleviating psychological distress and mitigating feelings of inadequacy and self-fragmentation in susceptible individuals who struggle with the rapid pace of modern life. From this perspective, substance use disorders (SUDs) and related psychopathologies can be viewed as a culture-bound phenomenon deeply entrenched in the core principles of contemporary fluidity. In addition to conventional psychoactive substances, nowadays more potent in terms of psychoactive principles, emerging concerns are directed towards novel psychoactive substances (NPSs), particularly due to the rapidly evolving and virtually boundless online market they inhabit [3,4,5,6,7,8]. The European Union (EU) has legally defined NPS as new narcotic or psychotropic substances, whether in pure form or preparations, which are not classified under the Single Convention on Narcotic Drugs of 1961 or the Convention on Psychotropic Substances of 1971 [9]. However, they present a public health risk potentially akin to the substances listed in these conventions. By 2017, the European Union had identified over 670 NPS, with 632 being identified post-2004 [10]. These substances often evade detection by health professionals due to the lack of reliable, evidence-based sources of information. The internet, including online forums, chat rooms, and blogs, has emerged as a primary platform for drug-related information dissemination, with an estimated 61% of young Europeans aged 15–24 citing it as a key source of drug-related information [11,12]. Within these platforms, users freely exchange their experiences with various substances, recommend sources, and discuss methods of administration. Vulnerable young individuals are particularly susceptible to aggressive marketing strategies employed by NPS vendors, such as enticing names, vibrant packaging, and free samples for trial, thereby increasing the likelihood of early substance exposure. Moreover, the largely unregulated nature of NPS contributes to their popularity, perpetuating the misconception of low associated risks [13,14]. Regrettably, comprehensive surveys regarding the prevalence of NPS diffusion remain scarce. Additionally, there is mounting evidence suggesting that NPS consumption often occurs unintentionally [15,16]. Increasing evidence underscores the significant psychiatric and physical risks associated with the consumption of NPS. Initially, phenethylamines and tryptamines constituted the most prevalent types of NPS. However, in recent years, cathinones, synthetic cannabinoids (SC), phencyclidine, and benzofurans have experienced a surge in popularity [17,18,19]. Several NPS have been directly or indirectly linked to severe adverse effects and fatalities. For instance, 2-DPMP and D2PM, synthetic stimulants belonging to the piperidine class, have exhibited both neuropsychiatric and cardiovascular toxicity, contributing to three deaths in August 2010; misuse of gamma-hydroxybutyrate (GHB) and gamma-butyrolactone (GBL) has been associated with over 150 fatalities in the UK between 1995 and 2013 [20,21]. More recently, the emergence of novel synthetic opioids presents a significant concern, contributing to a rising number of overdose deaths in both the United States and Canada [22,23,24]. The prevalence of novel psychoactive substances (NPS) is notable among adolescents and young adults, and it is particularly common among psychiatric patients who report experiencing psychotic symptoms at the onset of their illness. Furthermore, substantial evidence suggests that NPSs serve as a significant risk factor for violence and aggression among individuals diagnosed with major mental disorders [25,26,27,28].

Substance use frequently coexists with schizophrenia and other psychiatric conditions. The prevalence of substance use disorders (SUDs) among individuals with psychiatric illnesses is notably elevated, reaching up to 50%, a stark contrast to rates in the general population [29,30]. This high level of comorbidity poses significant challenges for clinicians and researchers in accurately distinguishing substance-induced psychopathology from primary psychiatric disorders within the context of concurrent SUDs [31,32]. According to the DSM-5, between 7% and 25% of individuals experiencing their first episode of psychosis exhibit symptoms of substance-induced psychosis (SIP). In a study conducted in Denmark, Norway, and Sweden, the annual incidence of SIP was estimated to be approximately 6.5 cases per 100,000 individuals, compared to 9.7 cases among those with a primary psychotic disorder (PPD) and comorbid substance use, and 24.1 cases among those with PPD alone. Individuals with chronic and heavy substance use, including cannabis, amphetamines, psychedelics, and cocaine, are at particularly elevated risk [31].

There is an increasing acknowledgment that substance use is linked to the onset of psychosis, which manifests during substance use and may persist even after withdrawal or cessation. Individuals experiencing SIP often seek urgent assistance, either by contacting law enforcement or ambulance services or by presenting themselves at emergency rooms, as they or their family and friends perceive an immediate need for intervention. Heavy and prolonged substance use can abruptly alter an individual’s perception and cognition, resulting in frightening experiences that may not be recognized as stemming from acute intoxication and necessitate urgent attention [33,34]. Some individuals may endure substance-induced psychotic symptoms for extended periods before seeking help. There is a dearth of research dedicated to differentiating substance-induced psychosis (SIP) from primary psychotic disorders (PPD), particularly in terms of elucidating potential divergent trajectories and outcomes [35,36,37]. Limited understanding exists regarding factors specifically associated with SIP, and there are doubts regarding the efficacy of current classification systems in effectively distinguishing between primary and substance-induced disorders. Notably, the concept of substances inducing transient psychotic states was initially discussed in studies dating back to the 1960s. Several experimental studies have since corroborated this notion, demonstrating drug effects that mirror both the positive symptoms (e.g., hallucinations, delusions, paranoia, and disorganized thinking) and negative symptoms (e.g., affective flattening, anhedonia, attentional and cognitive impairment) characteristic of schizophrenia [38,39,40]. Current major neurobiological theories of schizophrenia trace their roots back to the effects of various substances: the serotonergic model stemmed from observations of lysergic acid diethylamide (LSD) [41], the dopamine hypothesis emerged from studies on amphetamines, and the glutamatergic model was influenced by research on phencyclidine (PCP) and ketamine [42,43,44,45,46]. More recently, interest has grown in the role of endocannabinoids, spurred by the effects of cannabis [47,48,49].

A plethora of substances have been implicated in the onset of psychosis, leading to the inclusion of substance-induced psychosis (SIP) diagnoses in both the International Classification of Diseases 10 (ICD-10) [50] and the Diagnostic and Statistical Manual of Mental Disorders 5 (DSM-5) [51]. The term SIP was first introduced in the fourth edition of the DSM in 1994, and its diagnostic criteria have remained largely unchanged in the DSM-5. SIP is characterized by the emergence of psychotic symptoms, such as hallucinations and delusions, during or shortly after substance intoxication or withdrawal, with symptoms resolving within a specified timeframe. The DSM-5 stipulates that psychotic symptoms in SIP should be more severe than those expected during intoxication or withdrawal and should necessitate healthcare intervention. Additionally, there should be a lack of insight into hallucinations and delusions. However, it is notable that this definition does not explicitly encompass negative symptoms, potentially overlooking a significant aspect of clinical presentation. In the ICD-10, the time criterion for substance-induced psychosis (SIP) is slightly more stringent compared to the DSM-5, requiring partial resolution within one month and full resolution within six months. However, distinguishing between SIP and primary psychotic disorders (PPD remains a challenging diagnostic task in clinical practice. Misdiagnoses carry significant implications for clinical management, potentially leading to suboptimal follow-up and inappropriate treatment, with potential ramifications for prognosis.

In the DSM, a diagnosis of SIP hinges on the assumption that symptoms will dissipate following sustained abstinence. However, the literature indicates that diagnostic changes over time are common, with transition rates from SIP to PPD ranging from 25% to 50% [52,53]. Recent research suggests that SIP is associated with a substantial risk of transitioning to schizophrenia, particularly following cannabis-induced psychosis [54,55]. These high transition rates are attributed partly to the progression of the psychotic disorder (SIP evolving into PPD) and partly to the narrowed definition of SIP, which may predispose to misdiagnosis (SIP being diagnosed as schizophreniform disorder or psychosis NOS when schizophrenia criteria are not met) [56].

Neither the ICD-10 nor the DSM-5 allow for the coding of “persistent states” of SIP, despite mounting evidence suggesting that in some chronic users, psychotic symptoms can persist well beyond the specified timeframes. In such cases, the diagnosis is typically changed from SIP to PPD. However, the question remains unresolved regarding whether chronic substance use induces a long-lasting, clinically distinct psychotic syndrome, or if it precipitates a primary psychotic disorder such as schizophrenia. The challenge of differential diagnosis has hindered research into the unique psychopathology and future outcomes of individuals experiencing psychosis following substance use. These psychotic disorders may represent a distinct diagnostic category separate from primary psychotic disorders (PPD), exhibiting stability over time and possessing distinctive psychopathological characteristics.

### 1.2. De Clérambault’s Theory of Automatism

Gaëtan Gatian de Clérambault (1872–1934) was a French medical practitioner within the field of psychiatry. Beginning in 1905, he worked at the Special Infirmary for the Insane of the Paris Prefecture of Police, eventually assuming leadership from 1920 to 1934. This facility, with its 18 cells, served as a location for the temporary detention and psychiatric evaluation of individuals deemed potentially insane following police arrest [57]. Annually, around 2500–3000 individuals underwent psychiatric assessments in this service. De Clérambault was renowned for his rapid yet thorough clinical decision-making, with a focus on succinctly highlighting prominent symptoms rather than merely diagnosing specific disorders [58]. It is estimated that he authored between 13,000 and 15,000 certificates during his career [57]. In addition to his diagnostic role, de Clérambault also participated in teaching at the Special Infirmary for the Insane, primarily through public clinical presentations where he analyzed and discussed select cases [59]. Beginning with a brief paper in 1909, de Clérambault embarked on a series of writings concerning mental automatism throughout his career. These contributions proved highly influential, culminating in the 1927 conference of Aliénistes et Neurologistes de France, where the significance of de Clérambault’s research was highly recognized.

De Clérambault’s initial in-depth exploration of mental automatism dates back to 1920, followed by several papers throughout the decade [60,61,62,63,64,65,66,67,68]. By 1925, he defined it as “a certain clinical syndrome consisting of automatic phenomena in three registers: motor, sensory, and ideo-verbal” [68,69]. Here, ‘automatic phenomena’ refers to unexpected manifestations occurring within one’s body and/or mind, perceived as alien to the individual. Within specific functional registers (motor, sensory, and ideo-verbal), the person experiences passive submission to these parasitic elements, indicating a sense of misappropriation and interference [70,71,72,73].

These occurrences initially evoke surprise due to their spontaneous nature and mechanical expansion, yet they lack affective or thematic ideation, remaining largely neutral in nature [71]. What becomes apparent is the confusion and uncertainty stemming from disruptions in the process of ideation, resulting in an inability to comprehend the unfolding events [70]. This bewilderment does not arise from the specific mental contents encountered but rather from a fundamental disturbance in the formal structure of normal experience: “Mental automatism itself does not entail hostility. Initially, the constituent phenomena are emotionally neutral and lack thematic content” [66].

De Clérambault views mental automatism as a fundamental core process underlying various forms of mental illness [65,71]: “This syndrome encompasses all known types of hallucinations; however, the concept of mental automatism is broader than that of hallucination” [70]. While hallucinations are included, mental automatism denotes a general interference process affecting motor, sensory–affective, or ideo-verbal functioning, often observable in early psychotic states [66]. Non-hallucinatory mental automatism typically precedes the onset of hallucinations, although both phenomena may coexist [68]. Within the overarching process of interference, de Clérambault identified two categories of automatic phenomena: positive and negative. Put simply, automatism can manifest as the introduction of new elements into one’s functioning (positive mental automatism) or the deterioration of habitual contributing elements (negative mental automatism). The former constitutes intrusion phenomena, while the latter constitutes inhibition phenomena [69]. It provides an outline of specific clinical manifestations that de Clérambault identified as indicative of positive and negative automatic phenomena in motor, sensory–affective, or ideo-verbal functioning.

Throughout his work, de Clérambault [69] frequently made a distinction between minor automatism (petit automatisme) and major automatism (grand automatisme). Minor automatism refers to subtle positive and negative ideo-verbal phenomena that have yet to significantly impact an individual’s subjective functioning. Often, they go unnoticed, lacking specific content and remaining emotionally neutral. Major automatism involves sensory–affective and motor functioning, where prominent automatic phenomena, including ideo-verbal, disrupt the individual’s functioning, leading to feelings of confusion and overwhelm. Minor automatism often precedes major automatism, although this is not always the case. In some cases, thought echoes may emerge during the transition between these experiences [65]. De Clérambault’s approach diverged from conventional diagnostic systems, which focused primarily on specific symptoms like hallucinations and delusions, opting instead for a broader descriptive approach centered around mental automatism as a fundamental mechanism underlying various forms of psychosis. While traditional diagnostic methods aimed to provide an overall clinical picture of different psychotic conditions, de Clérambault believed that not all symptoms carried equal diagnostic weight. He sought to identify the elemental characteristic defining whether a condition was psychotic or not. According to his theory, patients experiencing hallucinations without automatic phenomena were not considered psychotic, whereas those experiencing automatic phenomena without descriptive symptoms like hallucinations and delusions were classified as psychotic. In developing his theory of mental automatism, de Clérambault relied on a descriptive framework that drew primarily from the works of his predecessors in French psychiatry. He particularly built upon Jules Baillarger’s ideas [74] regarding thought echoes and psychic hallucinations and Jules Séglas’s [75] work on verbal psychomotor hallucinations, both of which address the experience of imposed thoughts or speech. However, de Clérambault’s clinical syndrome uniquely grouped these phenomena with a wide range of other intrusive experiences that exerted similar effects on patients. Regarding subtypes of psychosis, de Clérambault associated mental automatism with various forms of paranoia, hypochondria, mania, melancholia, and hallucinatory psychoses. Despite these associations, he maintained a unified concept of psychosis, suggesting that similar automatic phenomena could lead to different types of delusions. For instance, cenesthetic sensations could give rise to hypochondria or delusions with mystical or persecutory themes. Central to de Clérambault’s work is his belief in a direct correspondence between observed clinical manifestations of mental automatism and the underlying neurological disturbances responsible for these automatic phenomena. He adhered strictly to an organicist perspective, positing that mental automatism resulted from negative reactions to factors such as infection, intoxication, or tumors. Since automatism itself followed a mechanistic logic and typically unfolded in a consistent pattern, he reasoned that its origins must also be mechanical rather than stemming from psychological conflicts, for instance [71]. Psychological explanations were deemed highly implausible by de Clérambault, leading him to dismiss them outright [72]. However, de Clérambault seemed to show little interest in the question of causation [76]. In fact, the concept of m‘echanism’ may have served more as a metaphorical construct [77].

### 1.3. Mental Automatism in Psychopathological Basis of Psychosis

According to his theory, mental automatism is the primary mechanism of psychosis, with other symptoms, such as delusions, being secondary reactions. He described delusions as necessary responses of an intact reasoning intellect to subconscious phenomena. Delusions are expansions of automatic phenomena that overwhelm and disturb the individual: ‘the intensity, unexpected nature, constancy, and strangeness of the sensation lead the subject to seek an external explanation’ [66] (p. 553). Delusional interpretations are cognitive responses that may eventually lead to the development of a ‘second,’ delusional personality [70]. Between the onset of automatic phenomena and the formation of a delusion, an ‘incubation period’ may occur, during which the initial experience of intrusion gradually permeates the patient’s broader mental life. This period is characterized by confusion due to conflicting thoughts and experiences: ‘an unexpected image arises, provoking an irrefutable thought; then it becomes haunting, provoking several contradictory thoughts’ [67].

De Clérambault assigned a different significance to the productive symptomatology of psychoses, such as the concept of hallucinations traditionally defined, based on a long 19th-century tradition, as “perceptions without objects”. Within the context of this still problematic differentiation, de Clérambault’s work, through his theory of mental automatism, seeks to acknowledge the clinical value of so-called psychic phenomena. He emphasizes the “non-sensory” nature of automatism, stating that “thoughts become foreign in the ordinary form of thought, that is, in an undifferentiated form rather than a defined sensory form. The undifferentiated form consists of a mixture of abstractions and tendencies, either without sensory elements or with vague and fragmentary multisensory elements”. He further adds that it is “an autonomous process often found in isolation and does not inherently imply any delusion, although a delusion may appear many years after its onset”. De Clérambault describes eidetic automatism as a disruption of thought that may include intrusive thoughts, imposed thoughts, and anticipation of thoughts. These phenomena constitute an “Echo of thought”, where the thought process is duplicated in time and space without the patient initially feeling particularly affected or persecuted, and often in the absence of delusion.

These characteristics are particularly relevant in clinical practice, as some patients exhibit similar phenomena that appear to exist independently of their underlying clinical condition. These phenomena can emerge at the onset or during the course of treatment, sometimes recognized by the patients as obstacles to therapy, and other times noticed only by the clinicians during interviews. What is particularly intriguing is the structural nature of these phenomena. In mental automatism, there is an element that seems to function autonomously, representing a prelude that can culminate in delusional crystallization.

### 1.4. The Interplay of Historical and Theoretical Perspectives in Mental Automatism

Although de Clérambault’s theories were dominant in French psychopathological schools, they were not embraced by German-speaking psychiatrists, where Bleulerian theories prevailed. However, his descriptions of subtle, micropsychotic phenomena—such as false recognitions, thought voids, perplexity, and verbal games—evoke Bleuler’s associative disorders as well as the subsequent descriptions of Schneider’s second-rank symptoms and Huber’s basic symptoms. This juxtaposition illustrates the interplay between different schools of thought, highlighting both the divergences and convergences in understanding psychotic phenomena. De Clérambault’s focus on the fine-grained, often overlooked aspects of psychotic experience contrasts with Bleuler’s broader categorization of associative disorders. While de Clérambault zeroed in on phenomena like false recognitions and thought voids, Bleuler categorized these under associative loosening. This difference in focus underscores a theoretical tension: de Clérambault’s detailed phenomenological approach versus Bleuler’s systemic categorization. This informs contemporary practice by encouraging a balanced approach that neither overlooks subtle phenomena nor loses sight of the broader clinical picture.

Karl Jaspers’ influence on Schneider’s criteria for diagnosing schizophrenia [78] is another critical link. Schneider’s second-rank symptoms, such as thought broadcasting and experiences of influence, bear resemblance to de Clérambault’s descriptions. The overlap suggests a foundational agreement on certain psychotic experiences across theoretical boundaries. However, the priority Schneider placed on first-rank symptoms reflects a shift towards more observable, less subjective phenomena. This interplay challenges modern clinicians to consider both observable and subtle symptoms in diagnosis and treatment.

Huber’s concept of basic symptoms [79] extends the conversation into the realm of early detection and intervention. His work on the pre-psychotic phase aligns with de Clérambault’s focus on micropsychotic phenomena, suggesting that early, subtle disturbances could precede full-blown psychosis. This alignment bridges historical theories with contemporary efforts in early intervention, challenging clinicians to refine their assessment tools to detect these early signs.

By examining these theories, we see a progression from detailed phenomenology to categorical diagnosis, and finally to early detection. This historical trajectory informs current practice by providing a multi-faceted framework for understanding and addressing psychosis. It challenges practitioners to integrate detailed phenomenological observation with structured diagnostic criteria and proactive intervention strategies.

## 2. Objectives

Through this historical analysis, we have understood that De Clérambault’s concept of automatism refers to a syndrome characterized by automatic phenomena in three domains: motor, sensory, and ideo-verbal. These phenomena manifest as unexpected and involuntary experiences within an individual’s body or mind, perceived as alien and intrusive. The scope of mental automatism includes:

Motor Automatism: Involuntary movements or actions that the individual feels are detached from their own volition. Sensory Automatism: Unanticipated sensory experiences, such as hallucinations, that occur independently of external stimuli. Ideo-Verbal Automatism: Intrusive thoughts or verbal expressions that seem to arise without the individual’s intentional effort.

These automatic phenomena are fundamental in understanding disruptions in normal cognitive and perceptual processes that can lead to psychosis. Initially, they typically lack emotional or thematic content but can evolve to significantly impact the individual’s mental state, leading to confusion and distress.

With this understanding, our work aims to revisit and expand upon de Clérambault’s theory of mental automatism through a phenomenological clinical investigation, assessing its relevance and application in the context of substance-induced psychoses (SIP). We aim to demonstrate how the psychopathological mechanisms in SIP differ formally and structurally from non-induced psychoses. Our goal is to rehabilitate de Clérambault’s intriguing theory of mental automatism, which we view as the foundational psychopathological mechanism driving the development of toxic psychoses. This work also draws upon theories from Janet and Jackson, as well as Bonhoeffer’s theory of exogenous psychoses.

Furthermore, we will explore the psychopathological mechanisms that facilitate the transition from acute to chronic psychosis. This includes identifying specific factors and conditions under which substance use leads to enduring psychotic states, beyond the immediate effects of intoxication or withdrawal. Based on these insights, we aim to synthesize a comprehensive framework for understanding SIP. This proposed framework will integrate the theory of mental automatism, the distinctions between induced and non-induced psychoses, and the mechanisms underlying the transition to chronic psychosis. By offering a cohesive model, we seek to contribute to the refinement of diagnostic criteria and therapeutic approaches for SIP, addressing existing gaps in clinical practice and research.

## 3. Results

### 3.1. Exploring Automatism in the Transition from Substance-Induced Psychosis to Persistent Psychosis

As mentioned above, we can deduce that the syndrome of mental automatism by de Clérambault, as is well known, was identified by the author between 1920 and 1926, as a fundamental component of the so-called “chronic hallucinatory delusions”. The subject affected by it perceives a series of elementary phenomena (motor, sensory, thought), which, despite occurring and manifesting in their internal experience, are effectively disconnected from their own will, thus appearing to them inexorable, uncontrollable, repetitive, and unrelated to anything else. It is as if within one’s own self, the person had objects that, regardless of their will, disrupt the order of space, changing position, appearing, and disappearing. These phenomena of automatism indeed have an autonomous, primitive, and neutral character, and, according to the author, they hold direct value as organic lesions. Specifically, the expression “automatisme mental” (Mental Automatism, *Geistsautomatismus*) denotes a very broad concept not uniformly understood, although its “organic” nature and “primary” character are unanimously acknowledged. From a psychopathological perspective, mental automatism—better specified with the attribute of “pathological”—is conceived both as a mechanism (of which the subject is unaware) of “liberation” of psychic elements that become autonomous from the ego and as a “feeling of automatism”, that is, conscious automatism albeit not “criticized” by the subject (who does not recognize its pathological nature). Drawing on the extensive Jacksonian theorizations, de Clérambault was convinced that intoxications could release activities normally held under the control of hierarchically superior inhibitory centers and thereby disengage them, allowing them to assume the character of automatism with seemingly spontaneous onset or at least unwanted. De Clérambault was therefore convinced that mental automatisms, due to their mechanistic nature, had a strong organic, hence biological, component. According to Jaspers [80], mental automatism constitutes a disturbance of ego consciousness, or, according to Schneider [78], of belonging to self. Essentially, it is a mechanistic, parasitic syndrome that profoundly disturbs the internal world of patients. De Clérambault deduced his observations on these cases while working at the Paris Prefecture [65], where he encountered individuals intoxicated by absinthe, ether, and chloral hydrate. For de Clérambault, therefore, automatisms were closely related to the organic substrate, namely what is defined as the biological impersonal [81]. The critical ego of the patient—this is a crucial aspect—helplessly witnesses the automatism, and, as de Clérambault asserts, the delusion overlays it.

### 3.2. The Dissociative Nucleus in the Exogenous Paradigm

De Clerambault’s seminal work delineates a dissociative nucleus that serves as a discerning observer of cascading automatic phenomena. This nucleus functions as the substratum for the genesis of secondary superstructures within the patient’s experiential domain, comprising manifestations such as delusional cogitation and sensory perturbations including tactile, visual, and auditory hallucinations. In essence, De Clerambault’s scholarship elucidates the intricate nexus between dissociation, automatic phenomena, and resultant psychopathological phenotypes. The aforementioned theoretical perspectives initiate a fresh discourse regarding the concept of “exogenous”, a term historically overshadowed by the prevailing endogenous paradigm. Initially, “endogenous” carried implications of hereditary influences and was frequently associated with degenerative processes, suggesting that certain disorders stemmed from inherent factors transmitted across generations, leading to progressive decline. The theories proposed by Morel [82] and Magnan [83] garnered support among German psychiatrists, highlighting degeneration as a pivotal element in endogenous disorders. However, Bonhoeffer [71] introduced a more elaborated perspective, contending that the clear differentiation between exogenous and endogenous factors was not straightforward. He acknowledged the complexity inherent in psychiatric disorders and the propensity for encountering mixed etiological factors, thereby blurring the distinction between exogenous and endogenous clinical features. Bonhoeffer posited, “We cannot definitively ascertain the ultimate nature of what is deemed endogenous… In practice, pure forms of exogenous and endogenous etiology are rare… A precise and exhaustive differentiation between exogenous clinical features and those classified as endogenous poses challenges”. Bonhoeffer’s perspective suggests that the differentiation between exogenous and endogenous factors may not always be unequivocal. This acknowledgment resonates with the intricate nature of mental health disorders, where diverse elements such as environmental stressors, trauma, genetic predisposition, and neurobiological mechanisms can converge to influence disorder onset. Such a viewpoint advocates for a more inclusive comprehension of the manifold influences on mental well-being, transcending rigid categorizations of exogenous versus endogenous factors. As research progresses, it becomes imperative to contemplate the interplay among genetic, environmental, and biological determinants in shaping the etiology and expression of psychiatric disorders. This holistic framework acknowledges the dynamic essence of mental health and underscores the necessity of considering a broad spectrum of factors in exploring the origins and trajectory of psychological conditions. Bonhoeffer’s initial assertions introduce the concept of external pathogenic agents and the formation of dissociative psychosis on a heteroplastic foundation, thereby disrupting the enduring discourse surrounding endogenous versus exogenous factors.

### 3.3. Automatism, Lysergic Psychoma and Toxic Psychosis

The theory of automatism by De Clerambault offers the possibility to explore the “dissociated” dimension of the mind and, in turn, the “parasitic” nature of the symptoms, which are structural aspects of substance-induced psychoses. In this context, the concept of lysergic psychoma presents significant potential for elucidating phenomena induced by various substances, transcending the scope of solely lysergic hallucinogens. Coined by Cargnello [39] and Callieri in 1963 [38], this term finds its conceptual roots in Hellpach’s definition and, notably, in the exogenous model formulated by Karl Bonhoeffer. Lysergic psychoma can be characterized as a psychopathological syndrome distinguished by the perception of an external entity intruding upon one’s cognitive processes. Clinically, it manifests with discernible symptoms impacting cognitive functions, mood regulation, and sensory experiences, particularly auditory and visual hallucinations, alterations in bodily sensations, and disruptions in spatial and temporal perception. Karl Bonhoeffer [71] integrated lysergic psychoma into the framework of hexogen psychosis models, wherein an external noxa (harmful agent) can precipitate and directly influence the onset of a comprehensive psychosis. Bonhoeffer initially documented this phenomenon while attending to alcoholic patients in Breslau, under the tutelage of Wernicke. His aim was to discern the underlying basis of functional dysfunction, identifying a hexogen complex capable of interacting with an endogenous background and culminating in complete psychosis. Bonhoeffer’s primary focus in his investigation lay on the compromised states of consciousness, spanning from disorientation and dream-like encounters to delirium and stupor. His exogenous model evolved through a dialectical interchange with Bleuler’s endogenous model, coinciding with Bleuler’s initial introduction of the term “schizophrenia”.

For a deeper understanding of the onset of exogenous lysergic psychoma, it is imperative to depict it as a distinctly egodystonic encounter. In this context, individuals become conscious of the presence of “unfamiliar or disturbed thoughts” within their own consciousness. The rational self can perceive and observe this experience as an unconventional event, beyond conscious control. This state is marked by vivid hallucinations, mainly of a visual and kinesthetic nature, alongside delusional perceptions. Occasionally, structured yet confined delusional thoughts may emerge. Despite cognitive disruption, the ego maintains a certain level of awareness and control over its role, often attempting to manage and contain the emerging psychoma. Usually, this psychosis remains self-limiting, aligning with the traits of an induced phenomenon driven by temporary pharmacodynamic effects induced by the substance. However, considering factors such as recurrence, high doses, and prolonged pharmacokinetic profiles of new compounds, an alternative scenario is worth contemplating.

When faced with a stable and recurrent abnormal encounter, the adaptive capacity and containment abilities of the thinking self might conceivably be overwhelmed. If atypical thoughts or aberrant perceptions assume a persistent nature, the ability to neutralize them could gradually wane. The thinking self could eventually lose its capacity to counter the encroachment of the psychoma, facilitating its intrusion into cognitive functioning. During this transition, the psychoma might extend its domain, becoming all-encompassing in the cognitive landscape. This pivotal moment could mark the onset of a psychotic experience characterized by widespread diffusion throughout the entire realm of consciousness. Chemical delusions, in this context, do not exhibit the typical features of delusional atmospheres, primary symptoms, or a pre-existing and persistent distortion of the structural dimensions of time, space, and intersubjectivity. To guard against the invasion of the lysergic psychoma, the integrity of the space–time continuum is safeguarded, ensuring that the shared realm of consciousness continues to provide meaning. If unfamiliar thoughts manage to infiltrate completely, it leads to the loss of the space–time dimension, leaving individuals in a deeply sorrowful state without the possibility of experiencing transcendence. Instead, individuals are characterized by a pronounced dissociative aspect, pronounced positive symptoms, the capacity for introspective analysis of these sensations, and a temporary disruption of time, space, and intersubjectivity, momentarily induced by the consumption of the substance itself. This psychopathological dimension encapsulates the core attributes of Bonhoeffer’s exogenous psychosis, a condition that historically has received comparatively less attention and recognition in contrast to Bleuler’s more prominent endogenous psychosis. The lysergic psychoma emerges as a significant contributor to the construction of meaning within these psychoses. De Clerambault’s concept of delirium, grounded in the notion of mental automatism, assumes a pivotal role. Within this framework, cognitive and emotional responses actively engage in mitigating the profound dissociative effects induced by substances, thereby serving as the primary driver in the genesis of delirium.

## 4. Discussion

At this point, we can approach two important reflections. The first is that the theory of mental automatism, originally conceived as the genesis of psychoses in general, can be applied to the development of substance-induced psychoses. The second point is that the transition to chronicity may be overdetermined after a basic biological lesion, triggered by external substances such as substances.

De Clérambault’s groundbreaking exploration of mental automatism between 1920 and 1926 [60,61,62,63,64,65] provided a detailed understanding of the genesis of psychoses. His concept of mental automatism and passivity, (forgotten in the European psychopathological tradition), characterized by the perception of uncontrollable and disconnected elementary phenomena, anticipated what Schneider coined as the term “*Gematch*” [78], challenging prevailing notions of psychosis by emphasizing its organic and mechanistic foundation. On the other hand, the intuition of mental automatism arises from the observation of thousands of intoxicated individuals, with psychosis triggered by substances such as absinthe and alcohol (in their acute phase), highlighting its biological nature. His observations underscored the intricate interplay between neurobiological mechanisms and psychopathological manifestations.

Moreover, de Clérambault’s delineation of a dissociative nucleus as the substrate for secondary psychopathological phenomena illuminated the complex nature of psychiatric disturbances. This perspective challenged traditional dichotomies between endogenous and exogenous factors and underscored the need for comprehensive frameworks that account for multifaceted influences on mental well-being. In light of de Clérambault’s contributions, Bonhoeffer advocated for a broader understanding of psychiatric disorders, one that transcended simplistic classifications and acknowledged the dynamic interplay between genetic predispositions, environmental stressors, and biological determinants. This shift in perspective prompted a reevaluation of traditional diagnostic paradigms, urging clinicians to adopt a more holistic approach to psychiatric assessment and treatment. Furthermore, the integration of de Clérambault’s concepts with contemporary perspectives on substance-induced psychoses, such as lysergic psychoma, enriched our understanding of the diverse manifestations of mental illness. By delineating substance-induced psychoses as egodystonic encounters marked by vivid hallucinations and disruptions in cognitive functioning, researchers gained insight into the complex interplay between pharmacodynamic effects and individual susceptibility factors. To better clinically summarize the psychotic progression triggered by substance-induced conditions, the frequent disturbances of cenesthesia and psychosensory experiences, especially visual, related to the use of hallucinogenic, dissociative, and excitatory substances, fall within the realm of mental automatism. At this juncture, the psychotic interpretive aspect emerges within a twilight atmosphere (marked by vivid visual hallucinations and extensive somatoesthetic phenomena). Subsequently, there is a progression of subcontinuous and continuous phenomena of mental automatism and states, including ideo-verbal, sensory–perceptual, and motor manifestations, until delusion fully manifests, often of a paranoid nature but always interconnected with the individual’s pre-existing personality dimensions. This delusion does not arise as revelation but as confirmation based on powerful sensory data. Finally, akin to states of automatism, the productive aspect appears egodystonic, as if the patient were a spectator of themselves, at least until chronic substance use sets in, and the development of recurrent psychosis episodes leaves no room for complete psychosis, undermining the previously maintained critical self.

Applying the aforementioned clinical considerations to a more clinical framework, the initial clinical elements that come into play in a patient with toxic psychosis are of an exclusively sensory–perceptual nature. These elements are closely associated with the prolonged use of substances with hallucinogenic, dissociative, or excitatory effects. During the twilight state [84,85] induced by these substances, it becomes much easier to fall prey to illusory–hallucinatory misperceptions, as the narrowing of the field of consciousness functions like a lens that focuses attention on details, thereby distorting them. This leads to sensations of itching, irritation, pain, and burning, which patients often attribute to a parasitic infestation of the body. These sensations are so intense that they globally alter the patient’s cenesthesia. The combination of sensory irritation and twilight consciousness organizes an experience of somatic passivity, which can be linked to mental automatism: the patient is aware, critical, and seeks help. They helplessly witness the progression of the psychosis.

Moreover, investigating the neural substrates involved in mental automatism can significantly improve our understanding of its underlying mechanisms. Recent studies highlight that analyzing the neural circuits and brain regions associated with mental automatism can offer valuable insights into its pathophysiology [86]. Neuroimaging techniques, such as functional MRI (fMRI) and PET scans, provide crucial data on the specific brain areas and neural pathways involved [87]. Key regions, like the prefrontal cortex and limbic system, play a vital role in the manifestation of mental automatism, thereby guiding more precise diagnostic criteria and therapeutic interventions [88,89].

These brain regions are essential for processing sensory information and regulating emotions, functions that are often disrupted by psychostimulant substances. This disruption creates an imbalance between the mind and body, initially presenting as subclinical irritation at the neurological interface. Over time, this irritation becomes conscious and phenomenological, manifesting in patients as delusional and hallucinatory structures [90].

## 5. Conclusions

Synthetic psychosis in new psychoactive substances and high-potency substance users is characterized by hallucinatory symptoms induced by a delusional interpretation. These patients are often misdiagnosed by mainstream psychiatry as having any mental disorder. According to official psychiatry, a paranoid schizophrenic overlaps with a heavy psychiatric diagnosis of the toxic etiopathogenesis of the disorder. As these individuals are introduced to ongoing psychiatric treatment, they might end up in a cycle of chronic psychiatric care. The use of neuroleptics can mask their symptoms to resemble those of traditional psychotics, hiding their underlying capabilities or strengths due to the side effects that mimic physical restraint. However, these patients have a more favorable outlook since their core sense of self remains intact, not eroded by the disintegration seen in other conditions. They do not suffer from loss of natural evidence [91], Bleulerian and Minkowskian autism, and do not undergo Kraepelinian deterioration processes [92,93,94]. This specific, atypical form of psychosis, prevalent among NPS and high-potency substance users, significantly affects social interactions and can lead to substance dependency or even suicidal tendencies. Properly identifying and treating this psychosis is crucial, as the future well-being of those struggling with addiction may hinge on it. This condition might be likened to an “Alice in Wonderland Syndrome” [84], where individuals experience psychotic symptoms without being inherently psychotic. This distinction underscores the critical difference between experiencing psychosis and being psychotic, suggesting that synthetic psychosis shares more similarities with biological or psycho-organic syndrome. By examining the core principles of psychopathology through a phenomenological perspective, we can gain a deeper understanding of a patient’s suffering and develop empathy for their situation. This approach encourages us to look beyond the superficial appearance of psychotic symptoms triggered by substance use, inviting us to explore the complex stories that the patient carries, waiting to be interpreted and narrated by the therapist. Unlike the traditional view of insanity that Michel Foucault [95] described, which has been predominantly associated with endogenous psychosis since the eighteenth century, substance-induced psychosis presents a distinct challenge. It emerges as a pivotal issue in the contemporary world, highlighting how psychiatric disorders evolve in response to societal changes and the development of new drugs. This shift signifies the dynamic nature of mental health issues as they adapt to the changing contexts of society and pharmacology.

### Future Perspectives

Mental automatism opens up intriguing reflections on the interdisciplinary dimension it traverses, intersecting neuroscientific hypotheses, phenomenological understanding, and experimental psychological applications. Insights from psychology contribute to our understanding of the cognitive and emotional processes involved in the perception and interpretation of automatic phenomena. Psychological theories on perception, cognition, and behavior can help explain how these automatic processes disrupt normal functioning and lead to psychotic symptoms. This is the essence of the predictive coding theory [96,97]. When the brain’s predictions do not match the actual sensory inputs, it has to adjust its expectations or alter perceptions, leading to the strange and involuntary experiences we know as automatisms. This modern theory aligns remarkably well with the early 20th-century observations of Gaëtan Gatian de Clérambault, who described automatic phenomena as occurring almost independently of conscious thought, though it places a stronger emphasis on the brain’s predictive capabilities. Another intriguing contemporary theory is the integrated information theory (IIT) [98], which explores the nature of consciousness itself. Imagine consciousness as a tapestry woven from countless threads of information, all intricately connected. When this integration falters, the tapestry unravels, leading to disjointed and automatic experiences. This perspective aligns with de Clérambault’s descriptions but adds a fresh, neurobiological viewpoint. IIT proposes that automatisms occur when the brain’s ability to integrate information is disrupted, resulting in a fragmented sense of self and perception.

Neuroscience offers a deeper understanding of the biological underpinnings of mental automatism. Neuroimaging studies can pinpoint the specific brain regions and neural circuits involved in these automatic phenomena, revealing how disruptions in neural connectivity and activity [99,100] result in the experiences described by Clérambault. These same areas are affected by substances of abuse, creating an interface between biological lesions and psychopathological symptoms that warrants further investigation.

Philosophy, particularly phenomenology, provides insights into the subjective experience of mental automatism. Phenomenological approaches emphasize the importance of understanding the lived experiences of individuals with psychosis, focusing on how they perceive and interpret their automatic phenomena. This perspective can enhance our empathy for patients and improve therapeutic approaches by addressing their subjective experiences.

Revisiting de Clérambault’s theory of mental automatism transcends mere exercises in psychopathology or historical exploration of the origins of psychosis. De Clérambault’s significant psychopathological contribution lies in identifying a biological substrate for psychotic symptoms, with a unique theoretical framework that emphasizes the patient’s ability to recognize and be aware of their own symptoms, contrasting with classic endogenous psychoses where passivity prevails. This approach is particularly relevant for toxic psychoses, aiding clinicians in better diagnosing conditions often misidentified and confused with other psychopathological forms, such as endogenous psychoses, which require different treatments and prognoses. The theory of mental automatism may have significant clinical applications and potential research developments in the field of toxic psychoses. By identifying the subtle manifestations of motor, sensory, and ideo-verbal automatism, which frequently appear in the early stages of substance-induced psychosis and are recognized by patients, healthcare providers can intervene earlier, implementing treatments that may prevent symptom escalation. Developing assessment scales for diagnosing patients with substance-induced psychotic episodes, particularly highlighting non-specific symptoms such as bodily sensations and hallucinations where the patient maintains awareness and self-coherence, is essential. Awareness of these psychopathological conditions promotes a therapeutic strategy that extends beyond antipsychotic pharmacology, especially since these patients often exhibit resistance to traditional dopaminergic antipsychotics. This strategy includes psychotherapeutic methods, such as interventions aimed at restoring normal sensory processing and cognitive filtering, which can effectively reduce automatic phenomena and alleviate psychotic symptoms. Educating patients and their families about mental automatism can demystify psychotic experiences and mitigate stigma. Understanding that these symptoms stem from automatic brain processes rather than personal failings can create a more supportive and therapeutic environment. Psychoeducational programs can incorporate information about mental automatism, aiding patients in recognizing and managing these symptoms more effectively.

Finally, several future research directions are essential, especially in this period marked by a progressive increase in psychopathological conditions induced by new-generation substances. Longitudinal studies tracking the progression of synthetic psychosis in NPS users over time will help identify critical factors influencing the transition from acute to chronic psychosis. Investigating the neural substrates of mental automatism and synthetic psychosis using advanced neuroimaging techniques can inform the development of targeted therapies. Formulating new therapeutic interventions to mitigate the automatic phenomena associated with synthetic psychosis, including both pharmacological treatments and novel psychotherapeutic approaches, is vital. Additionally, implementing prevention strategies to reduce NPS use, especially among vulnerable populations such as adolescents and individuals with pre-existing psychiatric conditions, is imperative. Promoting cross-disciplinary research that integrates insights from psychiatry, neuroscience, psychology, and pharmacology will provide a more comprehensive understanding of synthetic psychosis and its underlying mechanisms.

Advocating for policy changes and educational programs to address the risks associated with NPS use, increase public awareness, and provide accurate information about synthetic psychosis can help prevent its occurrence and improve early detection and treatment. By pursuing these research directions, we can enhance our understanding of synthetic psychosis and develop more effective strategies for its prevention and treatment, ultimately improving the lives of those affected by this condition.

## Data Availability

The data presented in this study are available on request from the corresponding author. The data are not publicly available due to privacy reasons.

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
