# Peer review of "Rethinking Mental Automatism: De Clérambault’s Theory in the Age of Novel Psychoactive Drugs: Psychotropic Effects and Synthetic Psychosis"

_healthcare, 2024, doi:10.3390/healthcare12121172_

Round 1

Reviewer 1 Report

Comments and Suggestions for Authors

30 April 2024

The review on the manuscript, titled ‘Rethinking Mental Automatism: De Clérambault’s Theory in the Age of Synthetic Psychoses’ by Ricci V et al., submitted to Healthcare.

Manuscript ID: healthcare-3007409

To Authors,

This commentary paper titled ‘Rethinking Mental Automatism: De Clérambault’s Theory in 2 the Age of Synthetic Psychoses’ delves into Substance-Induced Psychosis (SIP) diagnoses, comparing the criteria in the ICD-10 and DSM-5. It highlights the challenges in distinguishing SIP from Primary Psychotic Disorders (PPD) and the implications of misdiagnoses on clinical management. De Clérambault's concept of mental automatism as a core process in mental illness is detailed, discussing automatic phenomena and their impact on motor, sensory, and ideo-verbal functioning. The paper also discusses the interplay between exogenous and endogenous factors in psychiatric disorders, challenging traditional classifications. It concludes by emphasizing the importance of a holistic approach to psychiatric assessment and treatment, integrating De Clérambault's concepts with contemporary perspectives on substance-induced psychoses. The authors advocate for a deeper understanding of patients' experiences beyond superficial symptoms triggered by substance use, highlighting the dynamic nature of mental health issues in response to societal changes and pharmacology.

In general, I think the idea of this article is really interesting and the authors’ fascinating observations on this timely topic may be of interest to the readers of Healthcare. However, some comments, as well as some crucial evidence that should be included to support the author’s argumentation, needed to be addressed to improve the quality of the manuscript, its adequacy, and its readability prior to the publication in the present form.

Please consider the following comments:

·        The paper provides a thorough and detailed overview of the concept of mental automatism and its relationship to psychosis, covering both historical and theoretical aspects, and delving into the complexities of mental automatism, exploring its various manifestations, including motor, sensory, and ideo-verbal phenomena, and its role in the development of psychosis. Still, I believe that it could benefit from clearer organization and section headings to guide the reader through the complex ideas presented.

·        While the paper provides a detailed analysis of mental automatism, it would be helpful to include a clear definition of the concept and its scope at the outset, to provide a clearer understanding of the paper's focus and relevance.

·        While the authors integrated historical and theoretical perspectives, I believe that they should better discuss of how these perspectives inform and challenge each other.

·        I would recommend a more explicit discussion of the clinical implications of mental automatism. How does mental automatism inform our understanding of psychosis and its treatment? How can clinicians apply this concept in practice?

·        The paper concludes with a summary of the main points, but it could benefit from a more explicit discussion of future directions and potential avenues for research. This would provide a clearer sense of the paper's contributions and implications for the field.

·        Finally, I would suggest including clinical case studies or examples would help to illustrate the concept of mental automatism and its relevance to clinical practice, as well as addressing neural substrates to provide valuable insights into the underlying mechanisms of this phenomenon. Understanding how neural circuits and brain regions are involved in the manifestation of mental automatism could enhance our comprehension of the pathophysiology of psychotic disorders and substance-induced psychoses. By exploring the neural correlates of mental automatism, clinicians and researchers may gain a deeper understanding of the interplay between brain function and psychopathological manifestations, ultimately informing more targeted diagnostic criteria and therapeutic interventions [1-4].

·        I would suggest the authors to provide a more explicit discussion of the interdisciplinary approaches that are relevant to the study of mental automatism. How do insights from psychology, neuroscience, and philosophy inform our understanding of this concept?

I hope that, after careful revisions, the manuscript can meet the journal’s high standards for publication. I declare no conflict of interest regarding this manuscript.

Best regards,

Reviewer

1.      DOI: 10.3390/biomedicines12030613

2.      https://doi.org/10.3390/biomedicines12030574

3.      https://doi.org/10.1038/s41398-024-02737-x

4.      https://doi.org/10.17219/acem/185689

Author Response

 REVIEWER 1

The paper provides a thorough and detailed overview of the concept of mental automatism and its relationship to psychosis, covering both historical and theoretical aspects, and delving into the complexities of mental automatism, exploring its various manifestations, including motor, sensory, and ideo-verbal phenomena, and its role in the development of psychosis.

Still, I believe that it could benefit from clearer organization and section headings to guide the reader through the complex ideas presented.

Thank you for your thoughtful and constructive feedback on our paper. We are pleased to hear that you found our overview of mental automatism and its relationship to psychosis thorough and detailed, covering historical and theoretical aspects, as well as the various manifestations of mental automatism and its role in psychosis development. We appreciate your suggestion regarding the organization and section headings. We agree that a clearer structure could enhance the reader's ability to navigate the complex ideas presented. Consequently, we will revise the manuscript to include more defined section headings and improve the overall organization.

While the paper provides a detailed analysis of mental automatism, it would be helpful to include a clear definition of the concept and its scope at the outset, to provide a clearer understanding of the paper's focus and relevance.

Thank you for your insightful feedback on our manuscript.

We appreciate your suggestion to include a clear definition of mental automatism and its scope at the outset. We agree that this would provide a clearer understanding of the paper's focus and relevance from the beginning. Therefore, we will revise the OBJECTIVES chapter  to incorporate a precise definition of mental automatism and outline its scope. This will ensure that readers have a solid foundation for understanding the subsequent analysis.

We added as follows

Through this historical analysis, we have understood that De Clérambault's concept of automatism refers to a syndrome characterized by automatic phenomena in three domains: motor, sensory, and ideo-verbal. These phenomena manifest as unexpected and involuntary experiences within an individual's body or mind, perceived as alien and intrusive. The scope of mental automatism includes:

Motor Automatism: Involuntary movements or actions that the individual feels are detached from their own volition. Sensory Automatism: Unanticipated sensory experiences, such as hallucinations, that occur independently of external stimuli. Ideo-Verbal Automatism: Intrusive thoughts or verbal expressions that seem to arise without the individual's intentional effort.

These automatic phenomena are fundamental in understanding disruptions in normal cognitive and perceptual processes that can lead to psychosis. Initially, they typically lack emotional or thematic content but can evolve to significantly impact the individual's mental state, leading to confusion and distress.

With this understanding, our work aims to revisit and expand upon De Clérambault's theory of mental automatism through a phenomenological clinical investigation, assessing its relevance and application in the context of substance-induced psychoses (SIP). We aim to demonstrate how the psychopathological mechanisms in SIP differ formally and structurally from non-induced psychoses. Our goal is to rehabilitate De Clérambault's intriguing theory of mental automatism, which we view as the foundational psychopathological mechanism driving the development of toxic psychoses. This work also draws upon theories from Janet and Jackson, as well as Bonhoeffer's theory of exogenous psychoses.

Furthermore, we will explore the psychopathological mechanisms that facilitate the transition from acute to chronic psychosis. This includes identifying specific factors and conditions under which substance use leads to enduring psychotic states, beyond the immediate effects of intoxication or withdrawal. Based on these insights, we aim to synthesize a comprehensive framework for understanding SIP. This proposed framework will integrate the theory of mental automatism, the distinctions between induced and non-induced psychoses, and the mechanisms underlying the transition to chronic psychosis. By offering a cohesive model, we seek to contribute to the refinement of diagnostic criteria and therapeutic approaches for SIP, addressing existing gaps in clinical practice and research.

  • While the authors integrated historical and theoretical perspectives, I believe that they should better discuss of how these perspectives inform and challenge each other.

We agree that a more thorough examination of the interplay between historical and theoretical viewpoints will enrich the paper. Therefore, we will revise the relevant sections to:

We added as follows in the introduction chapter

Although his theories were dominant in French psychopathological schools, they were not embraced by German-speaking psychiatrists, where Bleulerian theories prevailed. However, his descriptions of subtle, micropsychotic phenomena (false recognitions, thought voids, perplexity, verbal games) evoke Bleuler's associative disorders as well as the subsequent descriptions of Schneider's second-rank symptoms and Huber's basic symptoms.

  • I would recommend a more explicit discussion of the clinical implications of mental automatism. How does mental automatism inform our understanding of psychosis and its treatment? How can clinicians apply this concept in practice?

Thank you for your insightful feedback. In response to your suggestion, we will revise the manuscript to include the following elements in the conclusion chapter. For instance, we create a subsection of conclusion chapter titled “future perspectives”:

Revisiting De Clerambault's theory of mental automatism is not merely an exercise in psychopathology or a study of the historical roots of psychosis. Reconsidering a long-forgotten theory can help clinicians better recognize substance-induced psychoses, whose diagnosis is often inadequate and confused with other psychopathological forms, particularly endogenous psychoses, which have entirely different treatments and prognoses. Recognizing the subtle signs of motor, sensory, and ideo-verbal automatism, which often present in early substance-induced psychosis and are noted by the patient, allows healthcare providers to intervene earlier, offering treatments that may prevent symptom escalation.

Awareness of these psychopathological conditions supports a therapeutic approach that is not solely based on antipsychotic pharmacology, especially as these patients often show resistance to classic dopaminergic antipsychotics, but also includes psychotherapeutic methods. For instance, interventions aimed at restoring normal sensory processing and cognitive filtering may be effective in reducing automatic phenomena and alleviating psychotic symptoms. Educating patients and their families about the nature of mental automatism can demystify psychotic experiences and reduce stigma. Understanding that these symptoms arise from automatic brain processes rather than personal failings can foster a more supportive and therapeutic environment. Psychoeducational programs can incorporate information about mental automatism, helping patients to recognize and cope with these symptoms more effectively

  • The paper concludes with a summary of the main points, but it could benefit from a more explicit discussion of future directions and potential avenues for research. This would provide a clearer sense of the paper's contributions and implications for the field.

Thank you for your constructive feedback  We appreciate your suggestion to include a more explicit discussion of future directions and potential avenues for research. Therefore, we will revise the conclusion to incorporate in the subsection “future perspectives” as follows

Finally, several future research directions are essential. Conducting longitudinal studies to track the progression of synthetic psychosis in NPS users over time will help identify critical factors influencing the transition from acute to chronic psychosis. Investigating the neural substrates of mental automatism and synthetic psychosis using advanced neuroimaging techniques can inform the development of targeted therapies. Developing new therapeutic interventions aimed at mitigating the automatic phenomena associated with synthetic psychosis, including both pharmacological treatments and novel psychotherapeutic approaches, is crucial. Designing and implementing prevention strategies to reduce the use of NPS, particularly among vulnerable populations such as adolescents and individuals with pre-existing psychiatric conditions, is also vital. Encouraging cross-disciplinary research that integrates insights from psychiatry, neuroscience, psychology, and pharmacology will provide a more comprehensive understanding of synthetic psychosis and its underlying mechanisms. Lastly, advocating for policy changes and educational programs to address the risks associated with NPS use, increasing public awareness, and providing accurate information about synthetic psychosis can help prevent its occurrence and improve early detection and treatment.

By pursuing these research directions, we can enhance our understanding of synthetic psychosis and develop more effective strategies for its prevention and treatment, ultimately improving the lives of those affected by this condition.

  • Finally, I would suggest including clinical case studies or examples would help to illustrate the concept of mental automatism and its relevance to clinical practice, as well as addressing neural substrates to provide valuable insights into the underlying mechanisms of this phenomenon. Understanding how neural circuits and brain regions are involved in the manifestation of mental automatism could enhance our comprehension of the pathophysiology of psychotic disorders and substance-induced psychoses. By exploring the neural correlates of mental automatism, clinicians and researchers may gain a deeper understanding of the interplay between brain function and psychopathological manifestations, ultimately informing more targeted diagnostic criteria and therapeutic interventions [1-4].

Thank you for your critical feedback. We have added an in-depth analysis of the biological correlates of this phenomenon mentioned these  references.

  1. DOI: 10.3390/biomedicines12030613
  2. https://doi.org/10.3390/biomedicines12030574
  3. https://doi.org/10.1038/s41398-024-02737-x
  4. https://doi.org/10.17219/acem/185689

Applying the aforementioned clinical considerations to a more clinical framework, the initial clinical elements that come into play in a patient with toxic psychosis are of an exclusively sensory-perceptual nature. These elements are closely associated with the prolonged use of substances with hallucinogenic, dissociative, or excitatory effects. During the twilight state (84) induced by these substances, it becomes much easier to fall prey to illusory-hallucinatory misperceptions, as the narrowing of the field of consciousness functions like a lens that focuses attention on details, thereby distorting them. This leads to sensations of itching, irritation, pain, and burning, which patients often attribute to a parasitic infestation of the body. These sensations are so intense that they globally alter the patient's cenesthesia. The combination of sensory irritation and twilight consciousness organizes an experience of somatic passivity, which can be linked to mental automatism: the patient is aware, critical, and seeks help. They helplessly witness the progression of the psychosis.

Moreover, investigating the neural substrates involved in mental automatism can significantly improve our understanding of its underlying mechanisms. Recent studies highlight that analyzing the neural circuits and brain regions associated with mental automatism can offer valuable insights into its pathophysiology (85). Neuroimaging techniques, such as functional MRI (fMRI) and PET scans, provide crucial data on the specific brain areas and neural pathways involved (86). Key regions, like the prefrontal cortex and limbic system, play a vital role in the manifestation of mental automatism, thereby guiding more precise diagnostic criteria and therapeutic interventions (87-88).

These brain regions are essential for processing sensory information and regulating emotions, functions that are often disrupted by psychostimulant substances. This disruption creates an imbalance between the mind and body, initially presenting as subclinical irritation at the neurological interface. Over time, this irritation becomes conscious and phenomenological, manifesting in patients as delusional and hallucinatory structures (89).

.

  I would suggest the authors to provide a more explicit discussion of the interdisciplinary approaches that are relevant to the study of mental automatism. How do insights from psychology, neuroscience, and philosophy inform our understanding of this concept?

Thank you for this suggestion. We addes as follows

Mental automatism opens up intriguing reflections on the interdisciplinary dimension it traverses, intersecting neuroscientific hypotheses, phenomenological understanding, and experimental psychological applications. Insights from psychology contribute to our understanding of the cognitive and emotional processes involved in the perception and interpretation of automatic phenomena. Psychological theories on perception, cognition, and behavior can help explain how these automatic processes disrupt normal functioning and lead to psychotic symptoms. This is the essence of the Predictive Coding Theory. When the brain's predictions don't match the actual sensory inputs, it has to adjust its expectations or alter perceptions, leading to the strange and involuntary experiences we know as automatisms. This modern theory aligns remarkably well with the early 20th-century observations of Gaëtan Gatian de Clérambault, who described automatic phenomena as occurring almost independently of conscious thought, though it places a stronger emphasis on the brain's predictive capabilities. Another fascinating contemporary theory is the Integrated Information Theory (IIT), which dives into the nature of consciousness itself. Picture consciousness as a tapestry woven from countless threads of information, all intricately connected. When this integration falters, the tapestry unravels, leading to disjointed and automatic experiences. This view resonates with De Clérambault's descriptions but brings a fresh, neurobiological perspective to the table. IIT suggests that automatisms arise when the brain's ability to integrate information is disrupted, causing a fragmented sense of self and perception.

Neuroscience offers a deeper understanding of the biological underpinnings of mental automatism. Neuroimaging studies can pinpoint the specific brain regions and neural circuits involved in these automatic phenomena, revealing how disruptions in neural connectivity and activity result in the experiences described by Clérambault. This neurobiological perspective can inform the development of targeted treatments aimed at addressing the underlying brain dysfunctions. Philosophy, particularly phenomenology, provides insights into the subjective experience of mental automatism. Phenomenological approaches stress the importance of understanding the lived experiences of individuals with psychosis, focusing on how they perceive and interpret their automatic phenomena. This perspective can enhance our empathy for patients and improve therapeutic approaches by addressing their subjective experiences.

I hope that, after careful revisions, the manuscript can meet the journal’s high standards for publication. I declare no conflict of interest regarding this manuscript.

Reviewer 2 Report

Comments and Suggestions for Authors

Thank you for submitting this manuscript. It was a pleasure to read, and I found the manuscript to be well worth reading. I myself am coming from a largely behavioral background, and reading about the work of de Clerambault was very stimulating.

At present, I am recommending an acceptance with minor revision. The present manuscript is, in my judgement, doing an excellent job of presenting a nuanced, technical argument of a means to understand the link between novel psychoactive substances and emergent psychosis. The authors' use of language is precise and technical, using all the correct terminology for the topic. Their argument is cohesive, blending the more theoretical work with a more modern understanding of biologically-based psychosis. And the manuscript is relatively short as well, perfect for this topic, and very well-cited.

My recommendations for changes have to do largely with readability and and your audience. First, the manuscript reads well, but I (and a couple of other readers who were interested in your topic) had difficulty following your section 2.1. I would advise adding in a couple of additional section titles (level 2 or 3) which describe the various points of your argument (automatism and psychosis, egodystonic similiarity, etc.) not to change your argument, but to help the reader identify key points that you are making. Second, I would recommend expanding your discussion, if feasible, on the neurobiological aspects of psychosis. This is less to make your manuscript neurobiological in nature and more to offer a clear connection to readers who may be interested in the mechanics behind the automatism. For example, when talking about the "dissociated" dimension of the mind, include brief mention of studies which have examined (qualitatively or quantitatively) measurement of these dimensions. You already have some ("Clinically, it manifests...."), just expand on this.

Thank you again for your submission. It was very stimulating to read, and I hope to see your revisions in the near future.

Author Response

REVIEWER 2

Thank you for submitting this manuscript. It was a pleasure to read, and I found the manuscript to be well worth reading. I myself am coming from a largely behavioral background, and reading about the work of de Clerambault was very stimulating.

At present, I am recommending an acceptance with minor revision. The present manuscript is, in my judgement, doing an excellent job of presenting a nuanced, technical argument of a means to understand the link between novel psychoactive substances and emergent psychosis. The authors' use of language is precise and technical, using all the correct terminology for the topic. Their argument is cohesive, blending the more theoretical work with a more modern understanding of biologically-based psychosis. And the manuscript is relatively short as well, perfect for this topic, and very well-cited.

Thank you for your kind words and positive feedback on our manuscript. We are delighted that you found our work stimulating and well-presented. We greatly appreciate your recommendation for acceptance with minor revisions.

We are glad that our efforts to present a nuanced and technical argument on the link between novel psychoactive substances and emergent psychosis were well received. We also appreciate your acknowledgment of our precise use of language, cohesive argumentation, and comprehensive citation.

My recommendations for changes have to do largely with readability and and your audience. First, the manuscript reads well, but I (and a couple of other readers who were interested in your topic) had difficulty following your section 2.1. I would advise adding in a couple of additional section titles (level 2 or 3) which describe the various points of your argument (automatism and psychosis, egodystonic similiarity, etc.) not to change your argument, but to help the reader identify key points that you are making.

We have revised the structure of the text by creating subsections that make the content more readable and less diffuse, in accordance with your suggestions.

 Second, I would recommend expanding your discussion, if feasible, on the neurobiological aspects of psychosis. This is less to make your manuscript neurobiological in nature and more to offer a clear connection to readers who may be interested in the mechanics behind the automatism. For example, when talking about the "dissociated" dimension of the mind, include brief mention of studies which have examined (qualitatively or quantitatively) measurement of these dimensions. You already have some ("Clinically, it manifests...."), just expand on this.

Thank you for your insightful suggestions for enhancing our manuscript. We appreciate your recommendation to expand the discussion on the neurobiological aspects of psychosis to provide a clearer connection for readers interested in the mechanics behind automatism.

In response to your feedback, we added as follows

Applying the aforementioned clinical considerations to a more clinical framework, the initial clinical elements that come into play in a patient with toxic psychosis are of an exclusively sensory-perceptual nature. These elements are closely associated with the prolonged use of substances with hallucinogenic, dissociative, or excitatory effects. During the twilight state (84) induced by these substances, it becomes much easier to fall prey to illusory-hallucinatory misperceptions, as the narrowing of the field of consciousness functions like a lens that focuses attention on details, thereby distorting them. This leads to sensations of itching, irritation, pain, and burning, which patients often attribute to a parasitic infestation of the body. These sensations are so intense that they globally alter the patient's cenesthesia. The combination of sensory irritation and twilight consciousness organizes an experience of somatic passivity, which can be linked to mental automatism: the patient is aware, critical, and seeks help. They helplessly witness the progression of the psychosis.

Moreover, investigating the neural substrates involved in mental automatism can significantly improve our understanding of its underlying mechanisms. Recent studies highlight that analyzing the neural circuits and brain regions associated with mental automatism can offer valuable insights into its pathophysiology (85). Neuroimaging techniques, such as functional MRI (fMRI) and PET scans, provide crucial data on the specific brain areas and neural pathways involved (86). Key regions, like the prefrontal cortex and limbic system, play a vital role in the manifestation of mental automatism, thereby guiding more precise diagnostic criteria and therapeutic interventions (87-88).

These brain regions are essential for processing sensory information and regulating emotions, functions that are often disrupted by psychostimulant substances. This disruption creates an imbalance between the mind and body, initially presenting as subclinical irritation at the neurological interface. Over time, this irritation becomes conscious and phenomenological, manifesting in patients as delusional and hallucinatory structures (89).

Thank you again for your submission. It was very stimulating to read, and I hope to see your revisions in the near future.

Reviewer 3 Report

Comments and Suggestions for Authors

This paper is a critical essay on mental automatisms in the light of novel psychoactive substances and classic psychopathology. Today, when scholars cope with the issue of different classification systems, reflecting on the history of psychiatric nosology is particularly important. My main suggestion is that authors reconsider the structure of the paper: “results,” “discussion,” and so on seem somewhat odd in this respect.  Please also define NPS even in the abstract. This is a rather heterogeneous class of substances with diverse biochemistry and mechanism of action. I also suggest that the authors include new cognitive theories of mental automatisms and compare these with the classic ones mentioned in the paper.

Author Response

REVIEWER 3

This paper is a critical essay on mental automatisms in the light of novel psychoactive substances and classic psychopathology. Today, when scholars cope with the issue of different classification systems, reflecting on the history of psychiatric nosology is particularly important.

My main suggestion is that authors reconsider the structure of the paper: “results,” “discussion,” and so on seem somewhat odd in this respect.

Thank you for this comment. We have revised the structure of the text by creating subsections that make the content more readable and less diffuse, in accordance with your suggestions.

Please also define NPS even in the abstract

Thank you.  we will revise the abstract to include a definition of NPS (Novel Psychoactive Substances).  We added as follows

The widespread use of Novel Psychoactive Substances (NPS)—defined as new narcotic or psychotropic substances not classified under the Single Convention on Narcotic Drugs of 1961 or the Convention on Psychotropic Substances of 1971—poses a significant challenge to contemporary mental health paradigms due to their impact on psychiatric health

This is a rather heterogeneous class of substances with diverse biochemistry and mechanism of action.

I also suggest that the authors include new cognitive theories of mental automatisms and compare these with the classic ones mentioned in the paper.

Thank you for this interesting suggestion. We added in the conclusion chapter as follows

Mental automatism opens up intriguing reflections on the interdisciplinary dimension it traverses, intersecting neuroscientific hypotheses, phenomenological understanding, and experimental psychological applications. Insights from psychology contribute to our understanding of the cognitive and emotional processes involved in the perception and interpretation of automatic phenomena. Psychological theories on perception, cognition, and behavior can help explain how these automatic processes disrupt normal functioning and lead to psychotic symptoms. This is the essence of the Predictive Coding Theory (95-96). When the brain's predictions don't match the actual sensory inputs, it has to adjust its expectations or alter perceptions, leading to the strange and involuntary experiences we know as automatisms. This modern theory aligns remarkably well with the early 20th-century observations of Gaëtan Gatian de Clérambault, who described automatic phenomena as occurring almost independently of conscious thought, though it places a stronger emphasis on the brain's predictive capabilities. Another fascinating contemporary theory is the Integrated Information Theory (IIT) (97), which dives into the nature of consciousness itself. Picture consciousness as a tapestry woven from countless threads of information, all intricately connected. When this integration falters, the tapestry unravels, leading to disjointed and automatic experiences. This view resonates with De Clérambault's descriptions but brings a fresh, neurobiological perspective to the table. IIT suggests that automatisms arise when the brain's ability to integrate information is disrupted, causing a fragmented sense of self and perception.

Neuroscience offers a deeper understanding of the biological underpinnings of mental automatism. Neuroimaging studies can pinpoint the specific brain regions and neural circuits involved in these automatic phenomena, revealing how disruptions in neural connectivity and activity result in the experiences described by Clérambault. This neurobiological perspective can inform the development of targeted treatments aimed at addressing the underlying brain dysfunctions. Philosophy, particularly phenomenology, provides insights into the subjective experience of mental automatism. Phenomenological approaches stress the importance of understanding the lived experiences of individuals with psychosis, focusing on how they perceive and interpret their automatic phenomena. This perspective can enhance our empathy for patients and improve therapeutic approaches by addressing their subjective experiences

Round 2

Reviewer 1 Report

Comments and Suggestions for Authors

3 June 2024

The 2nd review on the manuscript, titled ‘Rethinking Mental Automatism: De Clérambault’s Theory in the Age of Synthetic Psychoses’ by Ricci V et al., submitted to Healthcare.

Manuscript ID: healthcare-3007409

To Authors,

I am pleased that the authors have attempted to revise the manuscript based on the previous session. Nevertheless, the revisions have remained partial. Before publication, I respectfully request that the authors consider my comments and revise the manuscript to meet the high standards of the journal.

Comments:

1.      Keywords: Please include ten keywords from Medical Subject Headings (MeSH) in the title and the first two sentences of the abstract.

Best regards,

Reviewer

Author Response

Dear reviewer,

We have attempted to expand upon the concepts presented according to your suggestions. We hope that our responses are adequate and comprehensive. We remain available to address any specific points that may need further elaboration.

I am pleased that the authors have attempted to revise the manuscript based on the previous session. Nevertheless, the revisions have remained partial. Before publication, I respectfully request that the authors consider my comments and revise the manuscript to meet the high standards of the journal.

We thank the reviewer for the comment and have incorporated, as much as possible, the 10 MeSH terms in the title and the first two sentences of the abstract.

Therefore, we have changed the title in Rethinking Mental Automatism: De Clérambault’s Theory in the Age of Novel Psychoactive Drugs. Psychotropic Effects and Synthetic Psychosis.  We specified new words  in bold

Abstract: The widespread use of Novel Psychoactive Substances (NPS)—defined as new narcotic or psychotropic agents not classified under the Single Convention on Narcotic Drugs of 1961 or the Convention on Psychotropic Substances of 1971—poses a significant challenge to contemporary mental health paradigms due to their impact on psychiatric disorders. This study revisits and expands upon the theory of mental automatism as proposed by Gaëtan Gatian de Clérambault, aiming to elucidate the psychopathological mechanisms underlying Substance-Induced Psychoses (SIP) and their distinction from non-induced psychoses (Schizophrenia and related disorders).

  • The paper provides a thorough and detailed overview of the concept of mental automatism and its relationship to psychosis, covering both historical and theoretical aspects, and delving into the complexities of mental automatism, exploring its various manifestations, including motor, sensory, and ideo-verbal phenomena, and its role in the development of psychosis. Still, I believe that it could benefit from clearer organization and section headings to guide the reader through the complex ideas presented.

Thank you for your thoughtful and constructive feedback on our paper. We are pleased to hear that you found our comprehensive overview of mental automatism and its relationship to psychosis thorough, covering historical and theoretical aspects as well as the various manifestations of mental automatism and its role in psychosis development. We appreciate your suggestion regarding the organization and section headings. We agree that a clearer structure could enhance the reader's ability to navigate the complex ideas presented. Consequently, we will revise the manuscript to include more defined section headings and improve the overall organization.

We have restructured the introduction into four paragraphs:

  • Background on Substance-Induced Psychosis 1.2 De Clérambault's Theory of Automatism 1.3 Mental Automatism in Psychopathological Basis of Psychosis 1.4 The Interplay of Historical and Theoretical Perspectives in Mental Automatism

We have revised the objectives accordingly.

The results have been divided into three sections:

  1. Exploring Automatism in the Transition from Substance-Induced Psychosis to Persistent Psychosis
  2. The Dissociative Nucleus in the Exogenous Paradigm
  3. Automatism, lysergic psychoma and toxic psychosis

Additionally, we have added a subsection to the conclusion: 5.1 Future Perspectives

  • While the paper provides a detailed analysis of mental automatism, it would be helpful to include a clear definition of the concept and its scope at the outset, to provide a clearer understanding of the paper's focus and relevance.

Thank you for your insightful feedback on our manuscript.

Thank you for your insightful feedback on our manuscript. We appreciate your suggestion to include a clear definition of mental automatism and its scope at the outset. We agree that this would provide a clearer understanding of the paper's focus and relevance from the beginning. Therefore, we will revise the introduction with the addition of a new paragraph “Mental Automatism in Psychopathological Basis of Psychosis”. We added as follows

According to his theory, mental automatism is the primary mechanism of psychosis, with other symptoms, such as delusions, being secondary reactions. He described delusions as necessary responses of an intact reasoning intellect to subconscious phenomena. Delusions are expansions of automatic phenomena that overwhelm and disturb the individual: 'the intensity, unexpected nature, constancy, and strangeness of the sensation lead the subject to seek an external explanation' (de Clérambault, 1925, p. 533)(62). Delusional interpretations are cognitive responses that may eventually lead to the development of a 'second,' delusional personality (70). Between the onset of automatic phenomena and the formation of a delusion, an 'incubation period' may occur, during which the initial experience of intrusion gradually permeates the patient's broader mental life. This period is characterized by confusion due to conflicting thoughts and experiences: 'an unexpected image arises, provoking an irrefutable thought; then it becomes haunting, provoking several contradictory thoughts' (67).

De Clérambault assigned a different significance to the productive symptomatology of psychoses, such as the concept of hallucinations traditionally defined, based on a long 19th-century tradition, as "perceptions without objects." Within the context of this still problematic differentiation, De Clérambault's work, through his theory of mental automatism, seeks to acknowledge the clinical value of so-called psychic phenomena. He emphasizes the "non-sensory" nature of automatism, stating that "thoughts become foreign in the ordinary form of thought, that is, in an undifferentiated form rather than a defined sensory form. The undifferentiated form consists of a mixture of abstractions and tendencies, either without sensory elements or with vague and fragmentary multisensory elements." He further adds that it is "an autonomous process often found in isolation and does not inherently imply any delusion, although a delusion may appear many years after its onset".De Clérambault describes eidetic automatism as a disruption of thought that may include intrusive thoughts, imposed thoughts, and anticipation of thoughts. These phenomena constitute an "Echo of thought," where the thought process is duplicated in time and space without the patient initially feeling particularly affected or persecuted, and often in the absence of delusion.

These characteristics are particularly relevant in clinical practice, as some patients exhibit similar phenomena that appear to exist independently of their underlying clinical condition. These phenomena can emerge at the onset or during the course of treatment, sometimes recognized by the patients as obstacles to therapy, and other times noticed only by the clinicians during interviews. What is particularly intriguing is the structural nature of these phenomena. In mental automatism, there is an element that seems to function autonomously, representing a prelude that can culminate in delusional crystallization.

OBJECTIVES chapter to incorporate a precise definition of mental automatism and outline its scope. This will ensure that readers have a solid foundation for understanding the subsequent analysis. We added as follows:

We added as follows

Through this historical analysis, we have understood that De Clérambault's concept of automatism refers to a syndrome characterized by automatic phenomena in three domains: motor, sensory, and ideo-verbal. These phenomena manifest as unexpected and involuntary experiences within an individual's body or mind, perceived as alien and intrusive. The scope of mental automatism includes:

Motor Automatism: Involuntary movements or actions that the individual feels are detached from their own volition. Sensory Automatism: Unanticipated sensory experiences, such as hallucinations, that occur independently of external stimuli. Ideo-Verbal Automatism: Intrusive thoughts or verbal expressions that seem to arise without the individual's intentional effort.

These automatic phenomena are fundamental in understanding disruptions in normal cognitive and perceptual processes that can lead to psychosis. Initially, they typically lack emotional or thematic content but can evolve to significantly impact the individual's mental state, leading to confusion and distress.

With this understanding, our work aims to revisit and expand upon De Clérambault's theory of mental automatism through a phenomenological clinical investigation, assessing its relevance and application in the context of substance-induced psychoses (SIP). We aim to demonstrate how the psychopathological mechanisms in SIP differ formally and structurally from non-induced psychoses. Our goal is to rehabilitate De Clérambault's intriguing theory of mental automatism, which we view as the foundational psychopathological mechanism driving the development of toxic psychoses. This work also draws upon theories from Janet and Jackson, as well as Bonhoeffer's theory of exogenous psychoses.

Furthermore, we will explore the psychopathological mechanisms that facilitate the transition from acute to chronic psychosis. This includes identifying specific factors and conditions under which substance use leads to enduring psychotic states, beyond the immediate effects of intoxication or withdrawal. Based on these insights, we aim to synthesize a comprehensive framework for understanding SIP. This proposed framework will integrate the theory of mental automatism, the distinctions between induced and non-induced psychoses, and the mechanisms underlying the transition to chronic psychosis. By offering a cohesive model, we seek to contribute to the refinement of diagnostic criteria and therapeutic approaches for SIP, addressing existing gaps in clinical practice and research.

  • 3) While the authors integrated historical and theoretical perspectives, I believe that they should better discuss of how these perspectives inform and challenge each other.

We agree that a more thorough examination of the interplay between historical and theoretical viewpoints will enrich the paper. Therefore, we will revise the relevant sections to:

We have created another subsection in the Introduction

1.4 The Interplay of Historical and Theoretical Perspectives in Mental Automatism

Although De Clérambault's theories were dominant in French psychopathological schools, they were not embraced by German-speaking psychiatrists, where Bleulerian theories prevailed. However, his descriptions of subtle, micropsychotic phenomena—such as false recognitions, thought voids, perplexity, and verbal games—evoke Bleuler's associative disorders as well as the subsequent descriptions of Schneider's second-rank symptoms and Huber's basic symptoms. This juxtaposition illustrates the interplay between different schools of thought, highlighting both the divergences and convergences in understanding psychotic phenomena. De Clérambault's focus on the fine-grained, often overlooked aspects of psychotic experience contrasts with Bleuler's broader categorization of associative disorders. While De Clérambault zeroed in on phenomena like false recognitions and thought voids, Bleuler categorized these under associative loosening. This difference in focus underscores a theoretical tension: De Clérambault's detailed phenomenological approach versus Bleuler's systemic categorization. This informs contemporary practice by encouraging a balanced approach that neither overlooks subtle phenomena nor loses sight of the broader clinical picture.

Karl Jaspers' influence on Schneider's criteria for diagnosing schizophrenia (78) is another critical link. Schneider's second-rank symptoms, such as thought broadcasting and experiences of influence, bear resemblance to De Clérambault's descriptions. The overlap suggests a foundational agreement on certain psychotic experiences across theoretical boundaries. However, the priority Schneider placed on first-rank symptoms reflects a shift towards more observable, less subjective phenomena. This interplay challenges modern clinicians to consider both observable and subtle symptoms in diagnosis and treatment.

Huber's concept of basic symptoms (79) extends the conversation into the realm of early detection and intervention. His work on the pre-psychotic phase aligns with De Clérambault's focus on micropsychotic phenomena, suggesting that early, subtle disturbances could precede full-blown psychosis. This alignment bridges historical theories with contemporary efforts in early intervention, challenging clinicians to refine their assessment tools to detect these early signs.

By examining these theories, we see a progression from detailed phenomenology to categorical diagnosis, and finally to early detection. This historical trajectory informs current practice by providing a multi-faceted framework for understanding and addressing psychosis. It challenges practitioners to integrate detailed phenomenological observation with structured diagnostic criteria and proactive intervention strategies.

  • 4)  I would recommend a more explicit discussion of the clinical implications of mental automatism. How does mental automatism inform our understanding of psychosis and its treatment? How can clinicians apply this concept in practice?

Thank you for your insightful feedback. In response to your suggestion, we will revise the manuscript to include the following elements in the conclusion chapter. For instance, we create a subsection of the conclusion chapter titled "Future Perspectives" and we have sought to deepen the links between mental automatism and psychosis, especially from a clinical perspective. We added as follows

Revisiting De Clerambault's theory of mental automatism transcends mere exercises in psychopathology or historical exploration of the origins of psychosis. De Clerambault's significant psychopathological contribution lies in identifying a biological substrate for psychotic symptoms, with a unique theoretical framework that emphasizes the patient's ability to recognize and be aware of their own symptoms, contrasting with classic endogenous psychoses where passivity prevails. This approach is particularly relevant for toxic psychoses, aiding clinicians in better diagnosing conditions often misidentified and confused with other psychopathological forms, such as endogenous psychoses, which require different treatments and prognoses.

  • 5)    The paper concludes with a summary of the main points, but it could benefit from a more explicit discussion of future directions and potential avenues for research. This would provide a clearer sense of the paper's contributions and implications for the field.

Thank you for your constructive feedback  We added clinical applications and potential research developments of the theory of mental automatism, particularly in the context of toxic psychoses.  We will deepen the discussion on the links between mental automatism and psychosis, with a specific focus on clinical implications and future research directions.

The theory of mental automatism may have significant clinical applications and potential research developments in the field of toxic psychoses.. By identifying the subtle manifestations of motor, sensory, and ideo-verbal automatism, which frequently appear in the early stages of substance-induced psychosis and are recognized by patients, healthcare providers can intervene earlier, implementing treatments that may prevent symptom escalation. Developing assessment scales for diagnosing patients with substance-induced psychotic episodes, particularly highlighting non-specific symptoms such as bodily sensations and hallucinations where the patient maintains awareness and self-coherence, is essential. Awareness of these psychopathological conditions promotes a therapeutic strategy that extends beyond antipsychotic pharmacology, especially since these patients often exhibit resistance to traditional dopaminergic antipsychotics. This strategy includes psychotherapeutic methods, such as interventions aimed at restoring normal sensory processing and cognitive filtering, which can effectively reduce automatic phenomena and alleviate psychotic symptoms. Educating patients and their families about mental automatism can demystify psychotic experiences and mitigate stigma. Understanding that these symptoms stem from automatic brain processes rather than personal failings can create a more supportive and therapeutic environment. Psychoeducational programs can incorporate information about mental automatism, aiding patients in recognizing and managing these symptoms more effectively

  • 6) Finally, I would suggest including clinical case studies or examples would help to illustrate the concept of mental automatism and its relevance to clinical practice, as well as addressing neural substrates to provide valuable insights into the underlying mechanisms of this phenomenon. Understanding how neural circuits and brain regions are involved in the manifestation of mental automatism could enhance our comprehension of the pathophysiology of psychotic disorders and substance-induced psychoses. By exploring the neural correlates of mental automatism, clinicians and researchers may gain a deeper understanding of the interplay between brain function and psychopathological manifestations, ultimately informing more targeted diagnostic criteria and therapeutic interventions [1-4].

Thank you for your critical feedback. We have added an in-depth analysis of the biological correlates of this phenomenon mentioned these  references.

  1. DOI: 10.3390/biomedicines12030613
  2. https://doi.org/10.3390/biomedicines12030574
  3. https://doi.org/10.1038/s41398-024-02737-x
  4. https://doi.org/10.17219/acem/185689

Applying the aforementioned clinical considerations to a more clinical framework, the initial clinical elements that come into play in a patient with toxic psychosis are of an exclusively sensory-perceptual nature. These elements are closely associated with the prolonged use of substances with hallucinogenic, dissociative, or excitatory effects. During the twilight state (84) induced by these substances, it becomes much easier to fall prey to illusory-hallucinatory misperceptions, as the narrowing of the field of consciousness functions like a lens that focuses attention on details, thereby distorting them. This leads to sensations of itching, irritation, pain, and burning, which patients often attribute to a parasitic infestation of the body. These sensations are so intense that they globally alter the patient's cenesthesia. The combination of sensory irritation and twilight consciousness organizes an experience of somatic passivity, which can be linked to mental automatism: the patient is aware, critical, and seeks help. They helplessly witness the progression of the psychosis.

Moreover, investigating the neural substrates involved in mental automatism can significantly improve our understanding of its underlying mechanisms. Recent studies highlight that analyzing the neural circuits and brain regions associated with mental automatism can offer valuable insights into its pathophysiology (85). Neuroimaging techniques, such as functional MRI (fMRI) and PET scans, provide crucial data on the specific brain areas and neural pathways involved (86). Key regions, like the prefrontal cortex and limbic system, play a vital role in the manifestation of mental automatism, thereby guiding more precise diagnostic criteria and therapeutic interventions (87-88).

These brain regions are essential for processing sensory information and regulating emotions, functions that are often disrupted by psychostimulant substances. This disruption creates an imbalance between the mind and body, initially presenting as subclinical irritation at the neurological interface. Over time, this irritation becomes conscious and phenomenological, manifesting in patients as delusional and hallucinatory structures (89).

.

 7)  I would suggest the authors to provide a more explicit discussion of the interdisciplinary approaches that are relevant to the study of mental automatism. How do insights from psychology, neuroscience, and philosophy inform our understanding of this concept?

Thank you for your suggestion. We will provide a more explicit discussion of the interdisciplinary approaches that are relevant to the study of mental automatism. Insights from psychology, neuroscience, and philosophy each contribute uniquely to our understanding of this concept.

Mental automatism opens up intriguing reflections on the interdisciplinary dimension it traverses, intersecting neuroscientific hypotheses, phenomenological understanding, and experimental psychological applications. Insights from psychology contribute to our understanding of the cognitive and emotional processes involved in the perception and interpretation of automatic phenomena. Psychological theories on perception, cognition, and behavior can help explain how these automatic processes disrupt normal functioning and lead to psychotic symptoms. This is the essence of the Predictive Coding Theory. When the brain's predictions don't match the actual sensory inputs, it has to adjust its expectations or alter perceptions, leading to the strange and involuntary experiences we know as automatisms. This modern theory aligns remarkably well with the early 20th-century observations of Gaëtan Gatian de Clérambault, who described automatic phenomena as occurring almost independently of conscious thought, though it places a stronger emphasis on the brain's predictive capabilities. Another fascinating contemporary theory is the Integrated Information Theory (IIT), which dives into the nature of consciousness itself. Picture consciousness as a tapestry woven from countless threads of information, all intricately connected. When this integration falters, the tapestry unravels, leading to disjointed and automatic experiences. This view resonates with De Clérambault's descriptions but brings a fresh, neurobiological perspective to the table. IIT suggests that automatisms arise when the brain's ability to integrate information is disrupted, causing a fragmented sense of self and perception.

Neuroscience offers a deeper understanding of the biological underpinnings of mental automatism. Neuroimaging studies can pinpoint the specific brain regions and neural circuits involved in these automatic phenomena, revealing how disruptions in neural connectivity and activity (99-100) result in the experiences described by Clérambault. These same areas are affected by substances of abuse, creating an interface between biological lesions and psychopathological symptoms that warrants further investigation.

Philosophy, particularly phenomenology, provides insights into the subjective experience of mental automatism. Phenomenological approaches emphasize the importance of understanding the lived experiences of individuals with psychosis, focusing on how they perceive and interpret their automatic phenomena. This perspective can enhance our empathy for patients and improve therapeutic approaches by addressing their subjective experiences.

I hope that, after careful revisions, the manuscript can meet the journal’s high standards for publication. I declare no conflict of interest regarding this manuscript.